# Polymerized Molecular Allergoid Alt a1: Effective SCIT in Pediatric Asthma Patients

**DOI:** 10.3390/jcm14051528

**Published:** 2025-02-25

**Authors:** Giulia Brindisi, Alessandra Gori, Caterina Anania, Giovanna De Castro, Alberto Spalice, Lorenzo Loffredo, Alessandra Salvatori, Anna Maria Zicari

**Affiliations:** 1Department of Mother-Child, Urological Science, La Sapienza University, 00161 Rome, Italy; alessandra.gori85@gmail.com (A.G.); caterina.anania@uniroma1.it (C.A.); giovanna.decastro@uniroma1.it (G.D.C.); alberto.spalice@uniroma1.it (A.S.); annamaria.zicari@uniroma1.it (A.M.Z.); 2Department of Clinical Internal, Anesthesiological and Cardiovascular Sciences, Sapienza University of Rome, 00161 Rome, Italy; lorenzo.loffredo@uniroma1.it; 3Department of Woman, Child and General and Specialized Surgery, University of Campania “Luigi Vanvitelli”, 81100 Naples, Italy; alessandra.salvatori94@gmail.com

**Keywords:** Alternaria, allergic rhinitis, asthma, children, fractional exhaled nitric oxide testing, polymerized allergoids, subcutaneous allergen immunotherapy, spirometry

## Abstract

**Background**: Allergy to *Alternaria alternata* (Alt a), although often underdiagnosed, is a significant global health issue. In the allergen immunotherapy (AIT) field, novel therapeutic strategies are emerging, particularly with the advent of polymerized allergoids. This study aims to evaluate the efficacy of subcutaneous immunotherapy (SCIT) based on these innovative molecules in children with respiratory allergies, assessing clinical and functional parameters. **Methods**: We enrolled 42 patients aged between 6 and 16 years, all of whom had allergic rhinitis (AR) and concomitant asthma and all of whom were monosensitized to Alt a. Between December 2020 and December 2021, 17 patients initiated SCIT with Modigoid^®^ for Alt a1, while 25 patients continued with standard therapy. At the initial visit (T0), all the patients underwent nasal and bronchial evaluation, including exhaled nitric oxide (eFeNO) measurement and spirometry. The Asthma Control Test (ACT) was used to evaluate the control of asthma symptoms. Patients were followed up every 6 months, with a comprehensive re-evaluation at 24 months (T1) replicating the initial assessments. **Results**: After 24 months of SCIT with the new polymerized molecular allergoid Alt a1 (Modigoid^®^), children showed a statistically significant reduction in eFeNO levels, improved FEV1 values, and enhanced ACT scores. **Conclusions**: SCIT with the new molecular allergoid Alt a1 significantly improves functional parameters (FEV1 and eFeNO) and subjective asthma symptoms (ACT scores) in children with AR and objective asthma signs. This treatment represents an effective preventive strategy that can be used to halt the progression of the classic atopic march from AR to asthma and potentially reverse the atopic march.

## 1. Introduction

Allergies represent one of the most widespread chronic pathologies in the Western world, which is often referred to as experiencing an “allergic epidemic” [1].

Among allergic diseases, asthma is one of the most widespread, affecting an estimated 262 million people in 2019 and causing 455,000 deaths [2]. It affects children and adults and has grown in prevalence in recent years. For these reasons, and considering that even though several treatments are available, it cannot be cured and many patients have failed to achieve complete control of their symptoms, asthma is included in the WHO Global Action Plan for the Prevention and Control of NCDs and the United Nations 2030 Agenda for Sustainable Development [3]. Although numerous factors have been associated with an elevated susceptibility to asthma development, pinpointing a singular, direct cause remains challenging. Asthma risk increases with a family history of the condition and concurrent allergic diseases like eczema and rhinitis. Urbanization and related lifestyle factors are linked to higher asthma prevalence. Early-life factors, such as low birth weight, premature birth, tobacco-smoke exposure, exposure to air pollution, and viral infections, significantly impact lung development and elevate asthma risk. Exposure to environmental allergens and irritants, including indoor and outdoor pollutants, house dust mites, molds, occupational chemicals, and dust, as well as allergic rhinitis, increase the likelihood of developing asthma [4,5,6,7,8,9,10,11]. Perennial environmental allergens are implicated in severe AR and asthma symptoms, especially in children [12,13,14,15]. Alternaria alternata (Alt a.) should always be considered, as should dust mites [16].

Several studies have highlighted that higher concentrations of Alt a. spores correlate with a significant worsening of symptoms and an increase in the use of anti-asthma drugs, emergency room visits, and hospitalizations [14,17,18,19]. In Europe, the rate of sensitization to Alt a. reached 8.9%, but Spain reported a higher prevalence of up to 20%, with young children particularly affected [20]. In Italy, a dated multicenter study involving 2,942 patients reported a prevalence of 10.4%, with significant regional variability. The lowest prevalence was found in the north of Italy, and the highest was reported in the island areas (1.8% in Turin; 29.3% in Cagliari). The highest rate of Alt a. positivity was found in males aged 11–20 and 21–30. Among Alt a.-positive patients, 79.7% reported allergic rhinitis (AR), and 53.3% reported asthma alone or with other symptoms [21]. *Alternaria alternata* is a fundamental allergen affecting 12.9% of the population in the USA, where it is so common that it is considered the “American mite” [22].

The primary allergen produced by Alt a., known as Alt a1, triggers IgE antibody reactions in about 80% of patients allergic to Alternaria, demonstrating unique molecular characteristics alongside its significant allergenic impact. Thus, purified natural or recombinant allergens have been used in allergen-specific immunotherapy (AIT) for Alt a. allergy, which is the only therapeutic strategy that modifies the natural course of asthma and allergic rhinitis [23,24,25] by modulating the immune system [26,27,28]. However, this approach often causes adverse reactions when it is used to treat Alt a. allergy [28,29,30,31,32,33,34]. Considering the well-documented benefits of AIT, this concern has driven the development of new formulation methods for AIT to enhance safety, improve patient adherence, and ultimately, promote the use of this effective option for allergy treatment.

Therefore, a new polymerized molecular allergoid Alt a1 has been devised to reduce side effects and enhance clinical benefits. Chemical modification of allergens to create polymerized allergoids involves the formation of intermolecular bonds, leading to the creation of polymers with reduced IgE reactivity while maintaining immunogenicity. This process results in the masking of allergenic epitopes, making only surface epitopes accessible for interaction with IgE, particularly on mast cells. The reduced IgE reactivity compared to unmodified allergens ensures the improved tolerability and safety of AIT by decreasing the allergens’ ability to bind IgE while preserving their immunogenicity [35,36]. Modifying allergens into allergoids optimizes the balance between immunogenicity and allergenicity, enhancing the efficacy and safety of AIT for allergic patients [37].

To date, studies on SCIT with polymerized Alt a1 have been limited; one study conducted among adults examined subcutaneous immunotherapy (SCIT) with the new polymerized molecular allergoid Alt a1 [38]. No pediatric studies are known. Thus, in 2020, our research group conducted new research in pediatric patients affected by AR and asthma signs, evaluating respiratory nasal function and cytology [39]. The present study considers additional data derived from continued follow-up of the same patients, evaluating lower-airway function using spirometric parameters, nitric oxide exhaled fraction (eFeNO), subjective parameters such as the asthma control test (ACT), and recourse to standard medications for both the treatment and control groups.

## 2. Methods

Our study was conducted in a real-world clinical setting in which patients received subcutaneous immunotherapy (SCIT) with polymerized Alt a1 as part of routine clinical practice. The study was non-randomized, with patients self-selecting into either the AIT or standard therapy group based on their preference and physician consultation. The treatment regimen, follow-up schedule, and outcome assessments (FEV1, FeNO, ACT scores) were consistent with standard clinical care, without deviations from routine practice. While this design reflects real-world treatment decisions and adherence patterns, the study followed a structured data-collection protocol to ensure rigorous outcome evaluation.

Patients aged 6–18 with moderately persistent AR, mild intermittent asthma, and mono-sensitization to ALT a were enrolled at the Pediatric Allergy Unit of the Department of Mother Child Urological Science, Sapienza University of Rome. During routine clinical practice, patients were offered SCIT with Modigoid^®^ Alt a1 in addition to symptomatic therapy, following the EAACI recommendations for using SCIT with aeroallergens [27].

Standard therapy consisted of nasal corticosteroids (Beclometasone Dipropionate 100 mcg), one puff per nostril once a day, for AR symptoms, and inhaled corticosteroids and bronchodilator spray (Salbutamol spray 100 mcg) for asthma symptoms. All patients received a standard baseline dose of inhaled corticosteroids (ICS) to maintain symptom control, with dosing individualized based on body weight as per GINA pediatric guidelines. This standardized approach ensured comparable treatment conditions across study groups.

The control group included patients matched for gender, age, allergic sensitization, and allergic pathology who declined SCIT and continued only with standard therapy.

The study was performed according to the Declaration of Helsinki regarding biomedical research involving human subjects. It was approved by the ethics committee of Policlinico Umberto I Sapienza University of Rome (protocol number 0441/2023). Written informed consent was obtained from all enrolled patients’ parents or legal guardians.

### 2.1. Patients Selection

Inclusion criteria included a clinical history of exposure to Alt a., a positive skin prick test (SPT) result of ≥3 mm, and the presence of specific molecular IgE levels for Alt a1 on testing using Immuno-CAP.

Exclusion criteria included uncontrolled asthma, severe chronic respiratory diseases, and any previous or concurrent immunotherapy for aeroallergens.

SCIT with Modigoid^®^ Alt a1 was proposed during routine clinical practice, following the EAACI recommendations for administering SCIT with aeroallergens [40]. At the initial visit (T0), all patients underwent blood sampling and nasal-functionality testing, as detailed in a previous study published with the same patient cohort [39,41].

They also underwent exhaled nitric oxide (eFeNO) testing and basal and post-bronchodilation spirometry. Each patient was evaluated in an outpatient setting during regular follow-up visits every six months. After 24 months of subcutaneous immunotherapy (SCIT) (T1), the same procedures conducted at T0 were repeated for every patient.

No clinical diagnostic interventions performed on patients differed from the routine clinical practice. The Asthma Control Test (ACT) [10] assessed the control of asthma symptoms. The asthma diagnosis was made according to GINA guidelines [42]. All patients were asked if they had reduced their use of standard therapy over the 24-month study period.

### 2.2. Calculating the Spore Season

There are several methods for calculating the spore season; in this case, we relied on the data provided in the 335/2020 report on “Allergenic Pollen in Italy: Analysis of Trends 2010–2019” conducted by the Italian Institute for Environmental Protection and Research (ISPRA), in collaboration with the 21 Regional Environmental Agencies (ARPA) and Provincial Environmental Agencies (APPA), which together form the National System for Environmental Protection (SNPA). For each monitoring station, the period between the start and end of the pollen season is characterized by the constant presence of airborne allergenic pollen belonging to Alt a. This period, which provides a comprehensive temporal dimension of the phenomenon, is defined as the Allergenic Pollen Season (SPA). This synthetic indicator offers a general temporal dimension of airborne allergenic pollen, which is extremely useful not only in aerobiology but also for allergologists, who can use it to support proper therapeutic planning and efficacy evaluation.

### 2.3. Exhaled Nitric Oxide (eFeNO)

eFeNO was measured during quiet breathing using a sensor positioned in the mouth and connected to a fixed-flow sensor. The analysis was conducted with a Cosmed Quark NO breath device, following the American Thoracic Society and the European Respiratory Society [43] guidelines.

### 2.4. Forced Expiratory Volume in the 1st Second (FEV1)

FEV1 is the key parameter for detecting bronchial obstruction via spirometry. It was measured using a Cosmed Spirometer (Cosmed, Rome, Italy), following ATS/ERS guidelines. FEV1 was expressed as a percentage of the predicted normal values and adjusted for height, gender, and ethnicity. According to ATS/ERS procedures, each patient underwent a spirometry post-bronchodilatation with 100 mcg of salbutamol spray after basal spirometry [44].

### 2.5. Asthma Control Test (ACT)

This questionnaire includes four items for the child (with a score for each item ranging from 0 to 3) and three for their parents (with a score for each item ranging from 0 to 5). The level of asthma control over the past 4 weeks is indicated by the total sum of each score, which provides an overall index. Patients with controlled asthma achieved a total score of 20 or higher, while those with uncontrolled asthma scored below 20 [10].

### 2.6. Schedule of Modigoid

SCIT with purified polymerized Alt a1 (Modigoid^®^ Alt a1, ROXALL Medicina España S.A., Zamudio, Spain) required a rapid build-up of 0.2 mL (0.8 μg of Alt a1), followed by 0.3 mL (1.2 μg of Alt a1) after a15-min interval. A maintenance dose of 0.5 mL (2 μg of Alt a1) was administered monthly until the conclusion of the treatment for each patient. The treatment typically lasts around 3 years. Patients were re-evaluated every 6 months, and at the 24-month mark, all the same procedures performed at T0 were repeated.

### 2.7. Statistical Analysis

Statistical analysis was performed using IBM SPSS version 27.0 (SPSS, Chicago, IL, USA). A normality test was conducted for the continuous variables, represented by the mean and standard deviation (SD). Nominal and ordinal variables were represented by relative counts and frequencies. The assumed clinical values were compared for cases and controls at baseline (T0) and follow-up (T1), using the paired-samples *t*-test or the Wilcoxon signed-rank test for paired samples. The comparison between cases and controls at both T0 and T1 was conducted using the *t*-test for unpaired samples or the Mann–Whitney U test. The relationship between immunotherapy use and the reduced need for symptomatic drugs was examined using the chi-square test. In all cases, a *p* value ≤ 0.05 was considered statistically significant.

## 3. Results

Forty-two patients allergic to Alt a. agreed to participate in this study. Of these, 25 received only standard therapy (the control group), while 17 started Modigoid in addition to standard treatment. The characteristics of the study population are outlined in Table 1 below.

All the patients underwent basal and post-bronchodilatation spirometry and eFeNO and were asked to compile the ACT. All the enrolled patients presented with moderately persistent AR and mild intermittent asthma. After the descriptive analysis of the data, we investigated the significance of all considered parameters for the case and control groups (p intra-group) at T0 and T1 and between the case group and the control group (p inter-group) between T0 and T1 (Table 2).

In the analysis of the intragroup variability at T0, we found no statistical significance for any of the variables examined in either the cases or the controls. We observed a statistically significant difference (*p* < 0.001) in eFeNO between T0 and T1 in the case group and at T1 between the cases and controls; refer to the box plot (Figure 1) and Table 2 for details.

**Figure 1 jcm-14-01528-f001:**
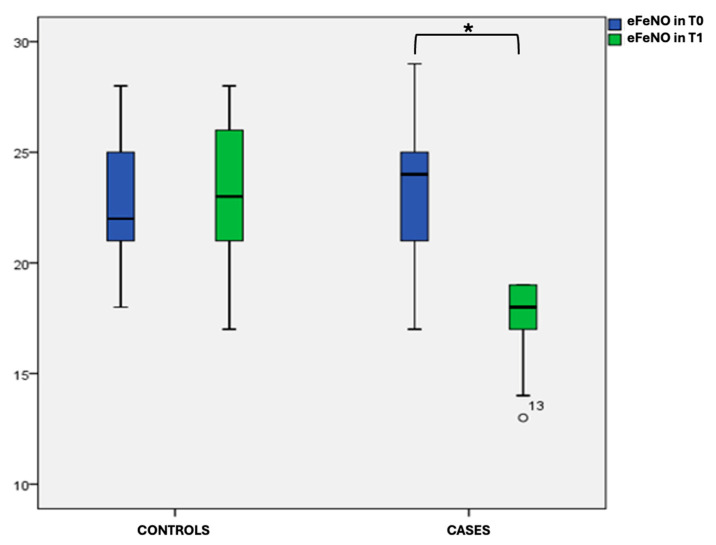
Box plot comparing eFeNO values at T0 and T1 in cases and controls.* statistically significant difference.

Additionally, all patients underwent baseline spirometry (FEV1 pre) and post-bronchodilation spirometry (FEV1 post). A significant difference was observed in the “FEV1-pre” and “FEV1-post” values between T0 and T1 within the case group (p-intragroup), as well as at T1 between case and control parameters (p-intergroup); refer to Figure 2A,B and Table 2.

**Figure 2 jcm-14-01528-f002:**
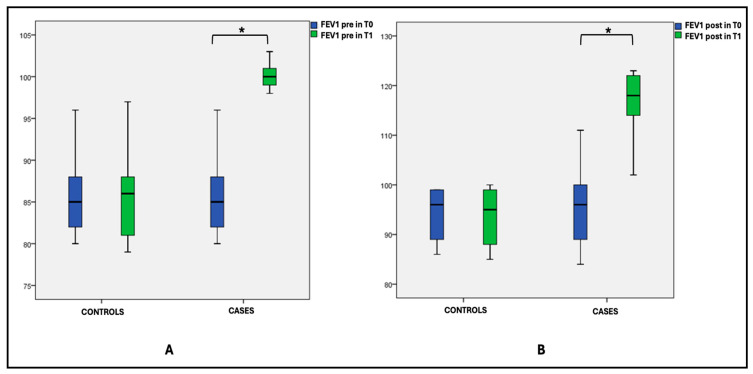
Box plot comparing the mean FEV1 values pre (**A**) and post bronchodilation (**B**) at T0 and T1 in cases and controls. * statistically significant difference. Also, in the analysis of the asthma control test (ACT), the comparison between ACT values at T0 and T1 in the case group showed a statistically significant difference (*p* < 0.001). The same significant values were found for ACT between cases and controls at T1 (*p* < 0.001), see Figure 3.

In addition, we evaluated the need for standard therapy for both the treatment and control groups. We observed a statistically significant reduction in the use of standard therapy for asthma symptoms in the treatment group compared with the controls at T1 (*p* < 0.001). During the entire duration of the study, we did not report any adverse reactions in our study population, illustrating the safety of this new Alt a1 SCIT.

## 4. Discussion

Allergy to Alt a is responsible for persistent and severe respiratory symptoms such as AR and allergic asthma [45,46].

The only treatment capable of modifying the immune system is allergen immunotherapy (AIT), which down-regulates the Th2 response, making it less reactive to single aeroallergens [40]. Moreover, in the era of precision medicine, AIT is becoming increasingly practical and capable of targeting a single allergen identified at the molecular level [47].

Advancing research aims to identify specific biomarkers that can predict the response to SCIT, which continues to be a prolonged treatment lasting at least three years.

Numerous studies have been conducted on Alt a. using the purified native allergen, which is not without risks of adverse reactions [11,21,26,27,28]. The significance of this new SCIT is related to the primary component of Alt a., known as Alt a1; this new version is chemically modified and polymerized to reduce the risk of adverse events. The molecular complexity of Alt a1, the major allergenic protein produced by the fungus *Alternaria alternata*, is multidimensional and involves unique structural and functional characteristics. The protein forms a dimeric ββ-barrel structure not found in other known proteins, a fact that highlights its uniqueness. This ββ-barrel structure consists of 11 ββ-strands that form a butterfly-like dimer linked by a disulfide bond. This structure is vital for the protein’s function and immunogenic properties, as it contains multiple IgE-binding epitopes that are crucial for allergic responses in sensitized individuals (Figure 4) [48].

Using allergoids for the new AIT decreases allergenicity while maintaining immunogenicity, thus offering improved safety and efficacy. In this way, it is possible to use higher doses of allergens in AIT preparation and shorten the process of tolerance development. In detail, Modigoid contains polymerized allergens, allowing higher, safer starting doses, fewer systemic reactions, and a consequent reduction in the number of injections needed, which is a strategic goal in pediatric patients [49]. The present study provides real-world evidence (RWE) on the effectiveness of SCIT with polymerized Alt a1 in pediatric patients with allergic asthma and allergic rhinitis. The observational nature of this study, with patient-driven treatment decisions and follow-up in a routine clinical setting, strengthens its applicability to daily practice.

Furthermore, objective (FEV1, FeNO) and subjective (ACT scores, medication use) clinical outcomes align with real-world asthma management. To further highlight this feature, the decision to begin patient enrollment and subsequently initiate therapy in December was carefully considered based on the aerobiological data from the previous year for the metropolitan area of Rome. This allowed us to schedule the biannual follow-up appointments to coincide with the beginning and end of the period of peak concentration of Alt a spores. The assessment of the use of medications included in the standard therapy was also scheduled for each patient during a specific period falling within the peak pollen season of Alt a. for the metropolitan area of Rome (RM 5 in Figure 5). Each botanical family has its own pollen season, during which the plants release significant amounts of anemophilous pollen into the atmosphere. Because seven plant families generate the vast majority of allergenic pollen monitored in Italy (Betulaceae, Corylaceae, Oleaceae, Cupressaceae-Taxaceae, Graminaceae/Poaceae, Compositae/Asteraceae, Urticaceae), we have seven different pollen seasons that follow and overlap each other continuously. For this reason, we selected patients residing in the metropolitan area of Rome who were monosensitized to Alt a. and selected a precise period.

Using this methodological approach, we observed that in the treatment group, children presented a statistically significant reduction in nasal FeNO levels, nasal eosinophils in cytology, total IgE, and specific IgE for Alt a1, as well as an improvement in NSS and nasal airflow during the anterior active rhinomanometry (AAR) [39,41].

The present study assessed the effects of Modigoid after 24 months of therapy on bronchial parameters, such as FEV1 (pre- and post-bronchodilation), through spirometry and events in the same cohort of patients. We observed a statistically significant increase in lung function in the treated group and a reduction in bronchial inflammation parameters. Additionally, we evaluated the subjective sensation of asthma symptoms using the ACT score, noting an improved score in the treatment group. To the best of our knowledge, this is the first study conducted in a pediatric population with the new allergoid Alt a1 that has assessed AIT efficacy in a post-registration or real-world context, incorporating objective methods such as spirometry and exhaled nitric oxide alongside subjective parameters like the ACT score.

In this study, following the excellent results found in the same group of patients regarding AR parameters, we discovered that effective control of AR led to excellent management of asthma symptoms and reduced allergic inflammation due to the continuity between the upper and lower airways [10,11]. Only one previous study had evaluated treatment with Modigoid, and it was conducted in adults; the results showed significant improvements in the AR and asthma symptoms of the adults enrolled after one year of therapy. Only one patient experienced a local adverse reaction; the treatment demonstrated complex beneficial effects with a significant improvement in symptoms and a good safety profile. However, no instrumental objective evaluation, nor any related to the patient’s subjective symptoms, was used in the study [38].

The other published studies on Alt a mainly relate to the AIT preparation with nAlt a1 or rAlt a1 commercial extract.

A comparative study of natural Alt a1 (nAlt a1) versus recombinant Alt a1 (rAlt a1) in 42 adults allergic to Alt a. showed no differences in IgE binding between nAlt a1 and rAlt a1. Both triggered a similar response in skin prick tests (SPTs) compared to the extract. The authors also demonstrated similar sensitivity and specificity, as measured by specific IgE levels, to nAlta1 or rAlta1. Thus, nAlt a1 and rAlta1 were helpful for the reliable diagnosis of Alt a. sensitization, showing performance comparable to that one of the Alt a. extract [50].

Morales et al. conducted original in vitro research comparing the new Alt a1 allergoid to its extract. The authors concluded that this new allergoid was safer than the extract, as it bound IgE less and induced tolerance through the shift of T cells towards Tregs (IL-10). No adverse events were recorded. Therefore, the new allergoid is considered an effective and safe therapy that stimulates cytokines through the synthesis of IgG antibodies, which can prevent the binding of IgE to the allergen [49].

A double-blind, randomized, placebo-controlled trial of subcutaneous immunotherapy with the purified major allergen Alt a1 was conducted by Tabar et al. to evaluate the efficacy and safety of two different doses of Alt a1 (0.2 vs. 0.37 mg of Alt a1 per dose) in patients with AR sensitized to Alt a (12–65 years of age) [33]. The authors found a statistically significant reduction in AR symptoms and the use of the drugs with the higher (0.37 mg) dose of Alt a1 compared to the placebo group after only one year of therapy. Reduced reactivity to SPTs, reduced IgE levels, and increased IgG4 levels were detected in both groups compared to the placebo group. No adverse events were reported. Therefore, the authors concluded that AIT with Alta1 was effective and safe, with the effect of reducing AR symptoms and drug consumption [33].

Another randomized double-blind trial conducted by Prieto et al. considered subjects between 9 and 60 years of age with AR and/or asthma who were sensitized to Alt a. and were randomized to receive placebo (*n* = 18) or SCIT with purified natural Alt a1 (n = 22) for 1 year [32]. At baseline and after 6 and 12 months of AIT, the authors evaluated bronchial responsiveness to adenosine 5′-monophosphate (AMP), methacholine, exhaled nitric oxide (eFeNO), exhaled breath condensate pH, and serum Alt a1-specific IgG4 antibodies [32]. They concluded that although AIT with purified nAlt a1 was well tolerated, no significant changes in AMP or methacholine responsiveness or improvements in inflammation markers in the exhaled air were detected, although there was an increase in IgG4 antibody levels [32].

Another randomized, double-blind, placebo-controlled, 3-year prospective study was conducted among patients (5–18 years of age) allergic only to Alt a. to evaluate the efficacy and safety of AIT with a standardized allergen extract. Fifty children and adolescents (25 girls; 5–18 years of age) with Alt a.-induced AR and/or bronchial asthma were randomly assigned to the case or placebo group. The primary endpoint was the value of the symptom medication score, evaluated subjectively. Secondary endpoints were the safety profile of the treatment, the patients’ quality of life, and the reactivity to allergen-specific nasal challenges. They demonstrated that the AIT with standardized Alt a. extract reduced symptoms of asthma and AR, and no severe side effects were reported [31].

The open-label, uncontrolled, multicenter, observational prospective study by L. Zapatero et al. enrolled patients between 4 and 16 years of age with persistent moderate to severe AR and/or mild to moderate asthma with sensitization to Alt a. only. A total of 99 patients were enrolled and treated with SCIT. After one year of follow-up, these patients were assessed for symptoms, medication use, quality of life, and asthma control, and a clear improvement in the parameters was observed. However, the limitations of this study are the lack of a control group and of the objective assessment of clinical improvements [20].

Additionally, original research conducted by Rodriguez et al., among 65 patients who started SCIT with Alt a1, found that immunotherapy ameliorated allergic symptoms and reduced relative drug consumption [51].

Only one multicenter and retrospective study was conducted on allergic asthma in a broad cohort of 582 patients with asthma from Alt a. They were divided into T2-high (n = 376) and T2-low (n = 206) groups with a threshold of 300 cells/μL in blood eosinophil count. Patients with T2-high asthma had relatively higher rates of inhaled corticosteroid (ICS) intake and a greater likelihood of having a positive family history compared to the T2-low group, as well as higher total IgE and sIgE for Alt a., with these values showing a significant positive correlation with the eosinophil count. The male predominance in patients with high Th2 asthma also appears in our sample, as already discussed in previous works on the same cohort of patients [52].

Thus, raising awareness of Alt a. sensitization in children as a risk factor for severe asthma and of the importance of an early diagnosis and intervention with SCIT lays the foundations for a precise therapeutic intervention tailored to the specific profile of the individual patient.

This is the first pilot study in which a subjective and objective evaluation was conducted among a pediatric population affected by allergic asthma who were undergoing SCIT treatment with a new polymerized allergoid.

The Asthma Control Test (ACT) improvement is a widely used and validated tool for assessing asthma control and provides valuable insight into the patient’s perception of symptom severity and disease management. However, as a subjective measure, it may be influenced by individual variability. To mitigate this limitation, we complemented subjective assessments with objective parameters such as spirometry (FEV1) and exhaled nitric oxide (eFeNO), ensuring a more comprehensive evaluation of treatment efficacy. This result is encouraging with regard the new frontier of molecular AIT, suggesting that these approaches can be used in the future, replacing the classic AIT approaches that use extracts and are often burdened by side effects.

AIT with the molecular allergoid Alt a1 is a new and powerfully effective approach for treating allergic asthma. Sensitization to Alt a prevents the progression of the atopic march and the development of molecular spreading, as well as sensitization to other aeroallergens. Furthermore, it is essential to educate pediatricians about the need to intervene as early as possible in clinical practice with AIT, especially in mild intermittent AR, to prevent disease progression and the development of AR comorbidities, such as allergic asthma.

## 5. Potential Sources of Bias and Additional Limitations in the Study

A primary concern is selection bias, as the study was not randomized and patients self-selected into either the SCIT (Modigoid^®^) group or the standard therapy group based on personal preference and physician consultation. This introduces variability, as patients choosing SCIT may have been more motivated or more health-conscious or may have had different baseline characteristics than those who opted for standard therapy alone. Additionally, the single-center design limits the generalizability of findings to other populations, geographic regions, or healthcare settings. Although the study groups were matched for age, gender, and allergic sensitization, other confounding factors such as lifestyle, environmental allergen exposure, and adherence to standard therapy may have influenced the outcomes. Furthermore, the study was open-label, meaning patients and clinicians were aware of treatment assignments, which could lead to observation bias, affecting symptom reporting and assessment. Another limitation is the small sample size, as the study included only 42 pediatric patients, with 17 in the SCIT group and 25 in the control group. A small sample size increases the risk of statistical variability, limiting the ability to detect small but clinically meaningful differences. Moreover, treatment adherence was assumed without objective monitoring, such as electronic medication tracking, and thus could have influenced treatment outcomes. The strict inclusion criteria, which required participants to be monosensitized to Alternaria alternata, limit the study’s applicability to the broader pediatric population, as polysensitization is more common in clinical practice. The study’s follow-up period was limited to 24 months, whereas SCIT is typically a three-year treatment. This means long-term efficacy, sustained immunological changes, and relapse rates remain unknown. While a statistically significant reduction in standard therapy use was reported, predefined criteria for reducing the use of corticosteroids or bronchodilators were not clearly outlined, potentially introducing variability in medication adjustments. Additionally, while follow-up intervals were standardized, true real-world studies often allow greater flexibility in visit schedules and treatment adjustments. Finally, electronic health records (EHR) or registry data were not utilized, although these could have provided broader real-world insights, including long-term adherence and effectiveness beyond controlled follow-up visits.

## 6. Strength

The strength points of this study are that it is the first study conducted on children of pediatric age to evaluate the effects of Modigoid from a subjective and objective point of view, considering respiratory lung function (FEV1) and lung inflammation (eFeNO).

## 7. Conclusions

This research provides compelling evidence supporting the use of subcutaneous immunotherapy (SCIT) with the new purified and polymerized allergoid Alt a1 in pediatric patients with allergic asthma, particularly in cases of Alt a sensitization. SCIT with Alt a1 significantly improved clinical and functional parameters, including reduced nasal and exhaled nitric oxide levels, reduced total IgE and specific IgE for Alt a1, improved nasal airflow, decreased nasal eosinophils, and enhanced FEV1 values. Moreover, SCIT with Alt a1 showed a preventive effect, blocking the progression of the atopic march from AR to asthma, with notable benefits observed from both subjective and objective clinical perspectives. These findings underscore the potential of early and targeted SCIT for mitigating the severity of asthma in children sensitized to Alt a, highlighting the importance of personalized immunotherapy approaches in managing allergic diseases in pediatric populations.

## Figures and Tables

**Figure 3 jcm-14-01528-f003:**
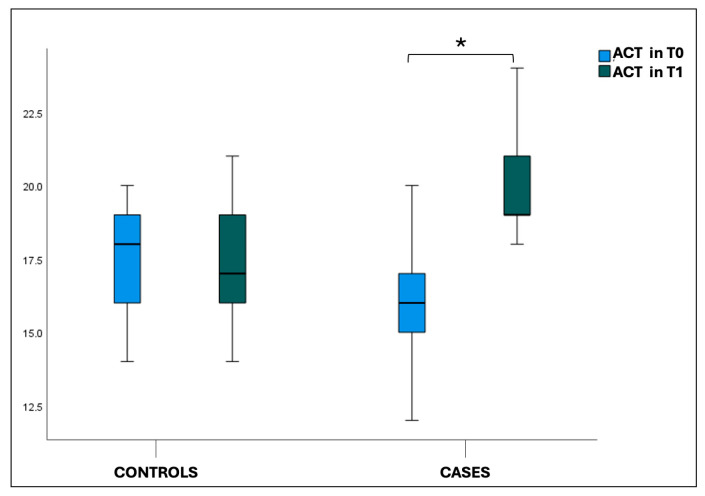
Box plot comparing ACT at T0 and T1 in cases and controls. * statistically significant difference.

**Figure 4 jcm-14-01528-f004:**
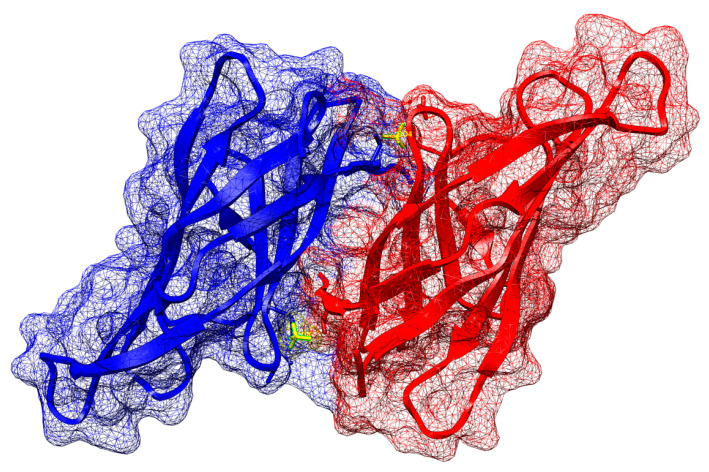
Alt a1 visualised with UCSF Chimera (Pettersen, E.F.et al. UCSF Chimera—A visualisation system for exploratory research and analysis. J. Comput. Chem. 2004) from R.P.D. Bank, «RCSB PDB-4AUD: Crystal structure of alternaria alternata majorallergen alt a 1». Available at https://www.rcsb.org/structure/4aud (accessed on 23 July 2024).

**Figure 5 jcm-14-01528-f005:**
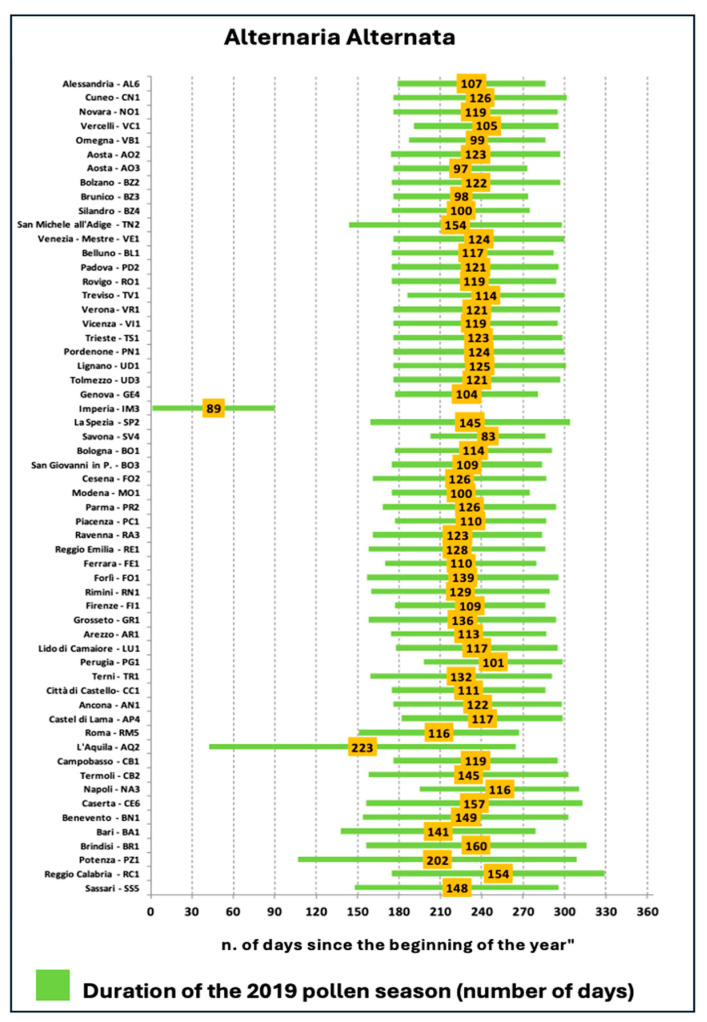
Alternaria allergenic season, 2019 (Adapted from ISPRA analysis on POLLnet data and Tor Vergata University, https://indicatoriambientali.isprambiente.it/it/qualita-dellaria/stagione-pollinica). Accessed on 15 December 2019.

**Table 1 jcm-14-01528-t001:** Characteristics of the study population. SD, standard deviation.

Characteristic	Control	Case	*p* Value
Numbers of patients (%)	25	17	
Male	21 (84%)	15 (88.2%)	
Age, mean ± SD, years	12.25 ± 6.26	12.55 ± 2.88	0.265
Weight, mean ± SD, Kg	43.19 ± 9.22	44.55 ± 8.23	0.672
Height, mean ± SD, cm	152.36 ± 10.80	154.68 ± 10.96	0.564
Allergic rhinitis,			
moderately persistent (%)	25 (100%)	17 (100%)	
Asthma,			
mild intermittent (%)	25 (100%)	17 (100%)	

**Table 2 jcm-14-01528-t002:** Intra-group and inter-group comparisons of the considered variables at baseline and after 24 months of therapy administration. FEV1: forced expiratory volume in the first second, expressed in liters (L); eFeN0: exhaled fraction of nitric oxide, expressed in parts per billion (ppb); ACT: asthma control test score (uncontrolled asthma < 20).

Characteristic	Value				*p* Value Intragroup
	Baseline		Post-Treatment (24 Months)		
	Mean ± SD	*p* Value Intergroup	Mean ± SD	*p* Value Intergroup	
**FEV1 pre-bronchodilatation** (L)					
Control	84.92 ± 3.94	*p* = 0.579	84.96 ± 4.35	*p* < 0.001	*p* = 0.846
Case	85.7 ± 5.16		100.23 ± 1.35		*p* < 0.001
**FEV1 post-bronchodilatation** (L)					
Control	93.84 ± 5.01	*p* = 0.385	93.89 ± 5.44	*p* < 0.001	*p* = 0.845
Case	95.59 ± 7.91		116.65 ± 5.95		*p* < 0.001
**eFeN0** (ppb)					
Control	22.76 ± 2.97	*p* = 0.573	22.88 ± 3.20	*p* < 0.001	*p* = 0.549
Case	23.29 ± 3.01		17.47 ± 1.90		*p* < 0.001
**ACT**					
Control	17.32 ± 1.7	*p* = 0.107	17.36 ± 1.89	*p* < 0.001	*p* = 0.841
Case	16.11 ±1.86		20 ± 1.8		*p* < 0.001

## Data Availability

The raw data supporting the conclusions of this article will be made available by the authors, without undue reservation.

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
