# Peer review of "Polymerized Molecular Allergoid Alt a1: Effective SCIT in Pediatric Asthma Patients"

_jcm, 2025, doi:10.3390/jcm14051528_

Round 1

Reviewer 1 Report

Comments and Suggestions for Authors

Thank you for the opportunity to review this study containing interesting results on the efficacy of modified, polymerized Alt a 1 allergoid with regard to asthma symptoms in pediatric subjects with allergic rhinitis. Please find below my comments and suggestions.

Major issues:

1.        The title in its current form may be somewhat misleading. The Authors in this study focused on asthma symptoms in subjects with allergic rhinitis, whereas the title of the article implies that rhinitis symptoms were the main focus. I suggest modification in a way that it highlights asthma symptoms and control assessment as the main focus of the study.

2.        I also suggest that in the first part of the article the real-world character of the study is clearly indicated.

3.        The introductory part, although desired and informative, is way too long, looking like a mini-review and not an introduction to the original article. Please consider selecting most relevant introductory information, leaving the rest out or transferring it to the discussion section.

4.        Since this is not a randomized study, what was the basis upon which the subjects where attributed to each group (AIT and no-AIT)? Was it patients’ (parents’) decision, doctors decision, other?

5.        Figure 4: Alternaria is not a pollen, therefore, usage of „pollen season” is not justified.

6.        Patients characteristics: are there any data available regarding the inhaled GCS dose in each group? If not avilale or not to be included, this should be addressed in the discussion as possible limitation of the study, considering influence of inhlaed GCS in airway inflammation and, therefore, intensity of symptom upon allergen exposure. Besides, did the results of the observation allow for lowering the inhalnt GCS dose during of aafter the study?

7.        Table 2: units should be provided for each parameter (FEV1, FeNO and so on). Please correct „pvalue” typo.

8.        Regarding statistics: the Authors indicate that normality test was performed. Please specify which test, and did all value sets passed normality test? This could be implied because mean (and not medians) are presented, but there were only 17 and 25 subjects in each group. Please make sure, that medians are used, in cases where values do not follow normal distribution.

9.        Lines 308-309. This fragment requires clarification. The Auhtros stated that only one study evaluated Modigoid efficacy. However, are there any RCTs with Modigoid basing on which the product was registered?

10.   In general, the discussion structure should be modified. Results of the present study are discussed in the context of previous ones, but the definite separation is suggested regarding which of the cited studies are RCTs and which are real-world studies, whenever applicable. This will present the current status and place of Modigoid more clearly.

Minor issues:

Alternaria Alternata (A.A.) is not commonly accepted as abbreviation. Should be in italics Alternaria alternata, abbreviated as Alt a or simply designed as Alternaria.

Layout should be modified: there many fragments where one sentence is one paragraph, not alway logically justified.

Line 299: please clarify what “reduction of even values” means exactly.

Throughout the article, the English language requires substantial amendments. There are also multiple layout errors, double spacings and very short paragraphs containing one sentence, which in most cases is not justified by the logical content of the respective fragments.

Comments on the Quality of English Language

Sometimes the manuscript does not read well. Formatiing amendment required as indicated. To be checked by a native speaker, if possible.

Author Response

Comment 1: The title in its current form may be somewhat misleading. The Authors in this study focused on asthma symptoms in subjects with allergic rhinitis, whereas the title of the article implies that rhinitis symptoms were the main focus. I suggest modification in a way that it highlights asthma symptoms and control assessment as the main focus of the study.

Response 1: We have modified the title into "Polymerized Molecular Allergoid Alt a1: Effective SCIT in Pediatric Asthma Patients."

Comment 2: I also suggest that in the first part of the article the real-world character of the study is clearly indicated.

Response 2: Dear Reviewer, We sincerely appreciate your valuable comments regarding the real-world nature of our study. Initially, we considered classifying it as a real-world study (RWS) due to its non-randomized design and the fact that patients received treatment within routine clinical practice. However, upon a more in-depth analysis of the defining characteristics of real-world studies, our response to the question "Is it a Real-World Study?" is "Partially", for the following reasons:

  1. Non-randomized and observational nature within a clinical setting: Our study was conducted in a real-world clinical environment, where patients autonomously decided whether to undergo SCIT with Modigoid® or continue with standard therapy alone. This reflects real-life decision-making in the management of pediatric allergic asthma, aligning with certain aspects of a real-world study.
  2. Limited generalizability: The strict inclusion criteria, selecting only patients mono-sensitized to Alternaria alternata, do not fully represent the broader allergic population, where polysensitization is common. Additionally, A true real-world study is often multi-center to enhance external validity and generalizability.
  1. Structured follow-up schedule: While follow-up visits were conducted as part of routine clinical practice, they followed a rigidly predefined schedule, which is less characteristic of true real-world studies, where follow-up intervals may vary.
  2. Absence of electronic health record (EHR) or registry-based data: True RWSs often leverage large-scale datasets, including national registries, administrative databases, or EHRs. In contrast, our study relied on prospectively collected data within a controlled study framework.

For all these reasons, we acknowledge that while our study possesses several real-world characteristics, it does not fully qualify as a real-world study. However, we have incorporated this reflection into the Methods, Discussion, and Limits sections, highlighting our study's strengths and limitations in this context. Particularly in the limit section, we have specified that the study was non-randomized, with patients self-selecting into either the AIT or standard therapy group based on their preference and physician consultation. However, in our country, allergen immunotherapy (AIT) is not reimbursed by the national health system and represents a significant financial burden for patients. As a result, despite having a clear indication for AIT, many patients and their families opt not to adhere to this treatment due to economic constraints. Consequently, these patients were allocated to the control group after informed consent. Therefore, the decision to undergo AIT was primarily influenced by financial considerations rather than clinical factors alone.

We added this paragraph to the Methods Section ( Page 2 line 45-52):

" This study was conducted in a real-world clinical setting, where patients received subcutaneous immunotherapy (SCIT) with polymerized Alt a1 as part of routine clinical practice. The study was non-randomized, with patients self-selecting into either the AIT or standard therapy group based on their preference and physician consultation. The treatment regimen, follow-up schedule, and outcome assessments (FEV1, FeNO, ACT scores) were consistent with standard clinical care, without deviations from routine practice. While this design reflects real-world treatment decisions and adherence patterns, the study followed a structured data collection protocol to ensure rigorous outcome evaluation."

We added this paragraph to the Discussion Section (Page 7 , Lines 8-12):

"This study provides real-world evidence (RWE) on the effectiveness of SCIT with polymerized Alt a1 in pediatric patients with allergic asthma and allergic rhinitis. The observational nature of this study, with patient-driven treatment decisions and follow-up in a routine clinical setting, strengthens its applicability to daily practice. Furthermore, the inclusion of both objective (FEV1, FeNO) and subjective (ACT scores, medication use) clinical outcomes aligns with real-world asthma management."

We have reorganized the Limit Section paragraph, now called “Potential Sources of Bias and Additional Limitations in the Study” acknowledging all previously noted.

Comment 3: The introductory part, although desired and informative, is way too long, looking like a mini-review and not an introduction to the original article. Please consider selecting most relevant introductory information, leaving the rest out or transferring it to the discussion section.

Response 3: We have substantially cut the introduction part, as requested.

Comment 4: Since this is not a randomized study, what was the basis upon which the subjects where attributed to each group (AIT and no-AIT)? Was it patients’ (parents’) decision, doctors decision, other?

Response 4: 

Thank you for your insightful question. As previously mentioned, the study was non-randomized, with patients self-selecting into either the AIT or standard therapy group based on their preference and physician consultation. However, in our country, allergen immunotherapy (AIT) is not reimbursed by the national health system and represents a significant financial burden for patients. As a result, despite having a clear indication for AIT, many patients and their families opt not to adhere to this treatment due to economic constraints. Consequently, these patients were allocated to the control group after informed consent. Therefore, the decision to undergo AIT was primarily influenced by financial considerations rather than clinical factors alone. We acknowledge this as a potential study limitation and have now clarified this point in the manuscript.

Comment 5:  Figure 4: Alternaria is not a pollen, therefore, usage of „pollen season” is not justified.

Response 5: Thank you for your observation. We acknowledge that Alternaria is a fungal spore rather than a pollen, and we have corrected the manuscript to ensure accurate terminology. We appreciate your attention to detail and your valuable feedback.

Comment 6: Patients characteristics: are there any data available regarding the inhaled GCS dose in each group? If not avilale or not to be included, this should be addressed in the discussion as possible limitation of the study, considering influence of inhlaed GCS in airway inflammation and, therefore, intensity of symptom upon allergen exposure. B) Besides, did the results of the observation allow for lowering the inhalnt GCS dose during of after the study?

Response 6: Thank you for your comment. All patients in the study received a standard baseline dose of inhaled corticosteroids (ICS) to maintain symptom control. Per standard pediatric practice, this dose was personalized based on the patient's weight. Given the pediatric population and the individualized dosing approach, specifying the exact dose for each patient in the manuscript was not feasible. However, we have added this sentence to clarify this aspect in the text: “All patients received a standard baseline dose of inhaled corticosteroids (ICS) to maintain symptom control, with dosing individualized based on body weight, under GINA pediatric guidelines. This standardized approach ensured comparable treatment conditions across study groups.” (Page 3, lines 3-6). 

Concerning the second question in the original paper on page 6, lines 1-3, we have previously specified that “In addition, we evaluated the need for standard therapy for both the treatment and control groups. We observed a statistically significant reduction in the use of standard therapy for asthma symptoms in the treatment group compared with the controls at T1 (p < 0.001)” .

Comment 7: Table 2: units should be provided for each parameter (FEV1, FeNO and so on). Please correct „pvalue” typo.

Response 7: Thank you for your observation. We have made the necessary corrections in the Table 2.

Comment 8: Regarding statistics: the Authors indicate that normality test was performed. Please specify which test, and did all value sets passed normality test? This could be implied because mean (and not medians) are presented, but there were only 17 and 25 subjects in each group. Please make sure, that medians are used, in cases where values do not follow normal distribution.

Response 8: Thank you for this comment. The normality test used is the Shapiro-Wilk Test, which in some cases did not allow the test sample to be considered perfectly normal (p < 0.05). However, even in those cases, the characteristics of the distribution associated with that sample - of unimodal type, with non-high skewness (-2 < skewness < +2 ), with tails at the two extremes of comparable length, with values of the mean and median not very different from each other - made it possible to assume negligible error in considering even that normal-type distribution. Therefore, it was decided to represent the summary values of the distribution associated with each quantity using Mean and Standard Deviation. In any case, the comparison between cases and controls, both at T0 and T1, was performed using both the T-Test for split samples and Mann Whitney's non-parametric U test, and in all cases the two tests always gave consistent results.

Comment 9: Lines 308-309. This fragment requires clarification. The Auhtros stated that only one study evaluated Modigoid efficacy. However, are there any RCTs with Modigoid basing on which the product was registered?

Response 9: Thank you for your comment. To clarify, our statement does not refer to the randomized controlled trials (RCTs) that may have supported the registration of Modigoid but rather to independent studies conducted after its approval. The study we mentioned specifically evaluates its efficacy in a real-world or post-registration context. We will revise the text to ensure this distinction is clear in this manner : “To the best of our knowledge, this is the first study, conducted in a pediatric population with the new allergoid Alt a1 that has evaluated AIT efficacy in a post-registration or real-world context, considering objective methods, such as spirometry and exhaled nitric oxide, in addition to subjective parameters, such as ACT score” (Page 8, lines 14-15).

Comment 10: In general, the discussion structure should be modified. Results of the present study are discussed in the context of previous ones, but the definite separation is suggested regarding which of the cited studies are RCTs and which are real-world studies, whenever applicable. This will present the current status and place of Modigoid more clearly.

Response 10: 

Thank you for your thoughtful suggestion. We agree that distinguishing between randomized controlled trials (RCTs) and real-world studies can provide additional clarity.  In our discussion, we aimed to present a comprehensive overview of the current evidence, integrating both types of studies to reflect the broader clinical context in which Modigoid is used for a more cohesive comparison of findings and better illustrates the clinical relevance of our results. Nevertheless, to address your concern, we have ensured that, where relevant, the study design of the cited literature is clearly indicated.

Minor issues:

Comment 1: Alternaria Alternata (A.A.) is not commonly accepted as abbreviation. Should be in italics Alternaria alternata, abbreviated as Alt a or simply designed as Alternaria.

Response 1: Thank you for your observation. We have changed A.A. into Alt a.

Comment 2: Layout should be modified: there many fragments where one sentence is one paragraph, not alway logically justified.

Response 2: Thank you for your observation. We acknowledge that some sections contain single-sentence paragraphs. However, this was done intentionally to enhance readability and emphasize key points. That said, we have carefully reviewed the text, and where appropriate, we have merged certain fragments to improve the logical flow and coherence of the discussion while preserving clarity.

Comment 3: Line 299: please clarify what “reduction of even values” means exactly.

Response 3: We modified the sentence in this manner:

“We found a statistically significant increase in lung function in the treated group and a reduction in bronchial inflammation  parameters” ( Page 8 , line 12).

Comment 4: Throughout the article, substantial amendments are required to the English language. There are also multiple layout errors, double spacings and very short paragraphs containing one sentence, which in most cases is not justified by the logical content of the respective fragments.

Response 4:Thank you for your valuable feedback. We have carefully reviewed the manuscript to address language-related concerns and have made necessary amendments to improve clarity and readability. Additionally, we have corrected any layout inconsistencies, including double spacings and unnecessarily short paragraphs, ensuring a more structured and coherent content presentation. We appreciate your attention to these details and believe the revised version significantly improves the overall quality of the manuscript.

Comment 5: Comments on the Quality of English Language. Sometimes the manuscript does not read well. Formatting amendment required as indicated. To be checked by a native speaker, if possible.

Response 5: Thank you for your feedback. We have carefully revised the manuscript, cutting as suggested by the second reviewer to improve readability and clarity, and addressing the formatting and language issues as indicated. Additionally, we have thoroughly reviewed the language to enhance fluency and coherence.

Reviewer 2 Report

Comments and Suggestions for Authors

This is an interesting paper, but redaction needs to be worked out to improve the flow of the paper. Several changes should be made.

.1. The assignment of participants to the intervention and control groups is described in the manuscript. However, it does not clarify whether randomization was performed. Please specify if the allocation of participants to groups was randomized. If randomization was applied, provide details about the randomization system or method used. 

2. Please clarify whether sample size calculations were performed for this study. If not, d provide sample size calculations, including power analysis, to ensure the study has adequate statistical power to detect significant differences.

3. Table 2  structure is functional but needs improvements in clarity and organization. We suggest adding a descriptive title, more informative column headers, and a legend explaining the abbreviations  (e.g., FEV1, eFeNO, ACT). Abecaus the table should be self-contained, allowing readers to understand its content without needing to refer to the main text

4. In Figures 1, 2 and 3 explain what does the asterisk mean.

5. In the discussion section, provide a more detailed analysis of the study design, including potential sources of bias and additional limitations. 

6. In the discussion comment whether randomization or blinding was applied, if there was no randomization comment if the absence of these can introduce selection and performance bias.

7. Commen  the impact of patient self-selection for treatment and the use of subjective measures such as the ACT score should be acknowledged. 

Regardin the paragraph from line 229 to  line 256 . The paragraph contains direct study findings and contextual information, affecting its clarity within the results section.  It should be split and changed into different places.

8. The sentences reporting study outcomes, such as the statistically significant reduction in the use of standard therapy and the absence of adverse reactions, should remain in the results section. 

9. Procedural details (e.g., the decision to begin patient enrollment in December and the scheduling of follow-up appointments based on the aerobiological data from Rome) should be moved to the methods section.

10. Broader contextual observations, such as the definition of pollen season and its relevance to therapeutic planning, should be moved to the discussion section. 

Author Response

Comment 1: The assignment of participants to the intervention and control groups is described in the manuscript. However, it does not clarify whether randomization was performed. Please specify if the allocation of participants to groups was randomized. If randomization was applied, provide details about the randomization system or method used. 

Response 1: Thank you for your insightful question. The study was non-randomized, with patients self-selecting into either the AIT or standard therapy group based on their preference and physician consultation. However, in our country, allergen immunotherapy (AIT) is not reimbursed by the national health system and represents a significant financial burden for patients. As a result, despite having a clear indication for AIT, many patients and their families opt not to adhere to this treatment due to economic constraints. Consequently, these patients were allocated to the control group after providing informed consent. The decision to undergo AIT was therefore primarily influenced by financial considerations rather than clinical factors alone. We acknowledge this as a potential limitation of the study and have now clarified this point in the manuscript.

Comment 2: Please clarify whether sample size calculations were performed for this study. If not, d provide sample size calculations, including power analysis, to ensure the study has adequate statistical power to detect significant differences.

Response 2: Thank you for your comment. As mentioned in the manuscript, to the best of our knowledge, this is the first study conducted in a pediatric population with the new allergoid Alt a1 that considers objective methods such as spirometry and exhaled nitric oxide in addition to subjective parameters such as ACT score. Additionally, our study was conducted in compliance with ethical guidelines, and our institutional ethical committee did not require a formal a priori sample size calculation (<30 patients for each group), given that this is an exploratory study aimed at assessing feasibility for a larger future trial. Furthermore, it is important to highlight that monosensitization to Alt a 1 is extremely rare, making it particularly challenging to recruit a larger sample size. However, the sample size calculation was performed by our group (data not exposed in the manuscript) based on expected effect sizes derived from previous studies on allergen immunotherapy for a better accuracy. Using a two-sample t-test power analysis (α = 0.05, power = 80%), we estimated that detecting significant differences in eFeNO and ACT scores would require approximately 14 patients per group, while FEV1 would require around 5 patients per group. Given our final sample size of 17 treated and 25 control patients, we believe our study is adequately powered for detecting meaningful clinical differences. Despite this, our findings provide valuable insights and serve as a foundation for future, larger-scale investigations.

Comment 3: Table 2  structure is functional but needs improvements in clarity and organization. We suggest adding a descriptive title, more informative column headers, and a legend explaining the abbreviations  (e.g., FEV1, eFeNO, ACT). Abecaus the table should be self-contained, allowing readers to understand its content without needing to refer to the main text.

Response 3 : We sincerely appreciate the reviewer’s insightful suggestion to improve the clarity and organization of Table 2. In response to this valuable feedback, we have made the following revisions:

  1. Added a more descriptive title to ensure that the purpose of the table is immediately clear to the reader.
  2. Refined the column headers to provide more precise and informative labels, improving readability.
  3. Included a legend beneath the table to define all abbreviations (e.g., FEV1 = Forced Expiratory Volume in 1 Second, eFeNO = Exhaled Fraction of Nitric Oxide, ACT = Asthma Control Test), making the table fully self-contained.

These modifications enhance the table’s clarity and ensure that readers can interpret the data independently without needing to refer to the main text.

We appreciate the reviewer’s valuable feedback and believe these revisions have significantly improved the presentation of Table 2.

 Comment 4: In Figures 1, 2 and 3 explain what does the asterisk mean.

Response 4: It means : * statistically significant difference

Comment 5: In the discussion section, provide a more detailed analysis of the study design, including potential sources of bias and additional limitations. 

Response 5: We have clarified in the discussion the study design characteristics and specified potential source of bias and additional limitation in the new section called “Potential Sources of Bias and Additional Limitations in the Study”

Comment 6: In the discussion comment whether randomization or blinding was applied, if there was no randomization comment if the absence of these can introduce selection and performance bias.

Response 6: As previously mentioned, we acknowledge this as a potential limitation of the study and have now clarified this point in the manuscript.

Comment 7: Comment  the impact of patient self-selection for treatment and the use of subjective measures such as the ACT score should be acknowledged. 

Response 7: We acknowledge that patient self-selection for treatment may introduce potential bias, as the decision to undergo SCIT was influenced by individual and family choices rather than random allocation. This factor could impact the generalizability of our findings, and we have now explicitly mentioned this as a limitation in the discussion.

Additionally, while the ACT score is a widely used and validated tool for assessing asthma control, we recognize that it remains a subjective measure dependent on patient perception. To mitigate this limitation, we complemented subjective assessments with objective parameters such as spirometry (FEV1) and exhaled nitric oxide (eFeNO), providing a more comprehensive evaluation of treatment efficacy. We have clarified this in the discussion section to ensure transparency regarding the study’s methodology and its potential limitations with this sentence in Page 10, Lines 3-to 7.

"The Asthma Control Test (ACT) is a widely used and validated tool for assessing asthma control, providing valuable insight into the patient's perception of symptom severity and disease management. However, as a subjective measure, it may be influenced by individual variability. To mitigate this limitation, we complemented subjective assessments with objective parameters such as spirometry (FEV1) and exhaled nitric oxide (eFeNO), ensuring a more comprehensive evaluation of treatment efficacy."

Regardin the paragraph from line 229 to  line 256 . The paragraph contains direct study findings and contextual information, affecting its clarity within the results section.  It should be split and changed into different places.

We appreciate the reviewer’s insightful suggestion to improve the clarity of the Results section by reorganizing the paragraph from lines 229 to 256. To enhance readability and ensure a more structured presentation of the study findings, we have made the following changes:

  1. Separated the direct study findings from the contextual background information.
  2. Moved the contextual information (background and methodological details) to the Methods section, ensuring that the Results section remains focused on study outcomes.
  3. Kept only relevant statistical findings in the Results section to maintain coherence with the overall data presentation.

Comment 8: The sentences reporting study outcomes, such as the statistically significant reduction in the use of standard therapy and the absence of adverse reactions, should remain in the results section. 

Response 8: Yes this part was already in the results section.

Comment 9: Procedural details (e.g., the decision to begin patient enrollment in December and the scheduling of follow-up appointments based on the aerobiological data from Rome) should be moved to the methods section.

Response 9: We have modified the text as suggested (Page 3, Lines 13 to 24)

Comment 10: Broader contextual observations, such as the definition of pollen season and its relevance to therapeutic planning, should be moved to the discussion section. 

Response 10: We have moved this paragraph to the discussion as suggested (Page 7 line 11-21). Thank you for this valuable recommendation, which has helped refine the manuscript’s structure.

Round 2

Reviewer 1 Report

Comments and Suggestions for Authors

The Authors have extensively addressed my comments, included some suggested amendments in to the text and provided rebuttal for other issues I raised. I agree with the arguments regarding features and characteristics of the real-world studies, that the Authors provide in their response. Also, the extensive explanation of statistics issues is sufficient. Not other issues to be indicated with regard to the manuscript.